# Influence of Aging Treatment Regimes on Microstructure and Mechanical Properties of Selective Laser Melted 17-4 PH Steel

**DOI:** 10.3390/mi14040871

**Published:** 2023-04-18

**Authors:** Dongdong Dong, Jiang Wang, Chaoyue Chen, Xuchang Tang, Yun Ye, Zhongming Ren, Shuo Yin, Zhenyu Yuan, Min Liu, Kesong Zhou

**Affiliations:** 1School of Materials and Energy, Guangdong University of Technology, Guangzhou 510006, Chinakszhou2004@163.com (K.Z.); 2National Engineering Laboratory of Modern Materials Surface Engineering Technology, Guangdong Provincial Key Laboratory of Modern Surface Engineering Technology, Institute of New Materials, Guangdong Academy of Sciences, Guangzhou 510651, China; 3State Key Laboratory of Advanced Special Steels, School of Materials Science and Engineering, Shanghai University, Shanghai 200444, China; cchen1@shu.edu.cn (C.C.);; 4Department of Mechanical and Manufacturing Engineering, Trinity College Dublin, University of Dublin, Parsons Building, D02 PN40 Dublin, Ireland; 5Wuhu Tianhang Equipment Technology Co., Ltd., Wuhu 241000, China

**Keywords:** selective laser melting, 17-4 PH steel, aging treatment regimes, microstructural evolution, mechanical properties

## Abstract

Aging is indispensable for balancing the strength and ductility of selective laser melted (SLM) precipitation hardening steels. This work investigated the influence of aging temperature and time on the microstructure and mechanical properties of SLM 17-4 PH steel. The 17-4 PH steel was fabricated by SLM under a protective argon atmosphere (99.99 vol.%), then the microstructure and phase composition after different aging treatments were characterized via different advanced material characterization techniques, and the mechanical properties were systematically compared. Coarse martensite laths were observed in the aged samples compared with the as-built ones, regardless of the aging time and temperature. Increasing the aging temperature resulted in a larger grain size of the martensite lath and precipitation. The aging treatment induced the formation of the austenite phase with a face-centered cubic (FCC) structure. With prolonged aging treatment, the volume fraction of the austenite phase increased, which agreed with the EBSD phase mappings. The ultimate tensile strength (UTS) and yield strength gradually increased with increasing aging times at 482 °C. The UTS reached its peak value after aging for 3 h at 482 °C, which was similar to the trend of microhardness (i.e., UTS = 1353.4 MPa). However, the ductility of the SLM 17-4 PH steel decreased rapidly after aging treatment. This work reveals the influence of heat treatment on SLM 17-4 steel and proposes an optimal heat-treatment regime for the SLM high-performance steels.

## 1. Introduction

As a typical additive manufacturing technique, selective laser melting (SLM) has gained great attention due to its unique advantages in fabricating sophisticated metallic parts [1]. Based on the powder bed fusion principle, a high-energy laser beam is used in the SLM technique to melt the powder feedstocks in a layer-by-layer sequence. Unlike conventional manufacturing methods, due to the micro-sized laser spot and fine particles, the SLM technique can fabricate metallic components with extremely high precision and geometrical complexity [2]. Additionally, the high-energy laser beam induces an ultra-high cooling rate during rapid solidification compared to conventional manufacturing techniques [3]. The laser absorptivity of the materials determines the quality of the components to a large extent [4]. Thus, the SLM-produced metallic components are commonly featured as equivalent or better mechanical properties [5]. To date, the SLM technique has been applied to the manufacturing of various kinds of metallic and composite materials, such as Ti alloys [6], Ni-based superalloys [7], Al-based alloys [8,9], Mg-based alloys [10,11], and Fe-based alloys [12].

Among all the steels, 17-4 PH (precipitation hardened) steel possesses an outstanding combination of high strength, excellent corrosion resistance, and favorable toughness [13,14,15]. Thus, 17-4 PH steel has been widely used in the tooling, molding, aerospace, and automotive industries [16,17,18]. Moreover, 17-4 PH steel is an excellent candidate for the SLM process due to its good weldability and wettability. Recently, some studies reported that the 17-4 PH steel fabricated by SLM achieved excellent mechanical properties by using optimal processing parameters [19,20]. Averyanova et al. [21] investigated the effect of the main processing parameters of SLM-fabricated 17-4 PH steel under the guidance of a fractional factorial approach. Although a complex function considering physical and geometrical parameters has been proposed, it merely describes the shapes of the fused tracks, while the mechanical properties based on the process parameters window were not reported. To further investigate the impacts of laser mode on microstructure features and mechanical properties, Ozsoy et al. [22] studied the impact of unique factors associated with pulsed laser processing using a design of experiments method. They successfully obtained SLM 17-4 PH steel samples with good mechanical performances (ultimate tensile strength = 899 MPa, elongation = 22.3%). However, the mechanical properties of 17-4 PH steel are still not adequate for some load-bearing applications.

Generally, the cyclic heating of deposited layers by subsequent laser movement in adjacent tracks and layers may trigger the precipitation of nano-scaled particles within the martensite matrix during the SLM process [23]. SLM-manufactured 17-4 PH steels are characterized by ultra-fine grains and a quantity of nanoparticles due to the intrinsic heat treatment (IHT) [24,25]. This IHT effect may also result in excellent strength for precipitation hardening steels such as 17-4 PH or maraging steel 300 (MS300) in the as-built condition [26]. However, limited improvement in the IHT effect means that SLM-manufactured 17-4 PH steels in as-built conditions cannot reach the standard of mechanical properties due to insufficient precipitation time.

Thus, post heat-treatments, such as solution and aging processes, are indispensable for 17-4 PH steels. The highly dispersed nano-scale soft Cu-rich particles induced by the post-heat treatments significantly improve the steel strength via the Orowan strengthening mechanism [27]. Many studies have demonstrated the significance of aging treatments for conventionally manufactured 17-4 PH steels. As such, inspired by the traditional heat treatments, heat-treated 17-4 PH steels are gradually becoming popular within the SLM community. Rafi et al. [28] reported that post-heat treatments such as solution and aging processes ameliorate the mechanical properties of SLM 17-4 PH steel. However, suitable post-heat treatments for SLM 17-4 PH steel samples were not summarized. Pasebani et al. [29] found that a high-temperature solution treatment would deteriorate the mechanical performance due to the coarsened martensite microstructure.

Most studies have focused on the effects of powder preparation methods, volumetric energy densities, and solution treatments on the microstructure and mechanical performance of 17-4 PH steel. However, systematic investigations on the direct aging treatment parameters for SLM 17-4 PH steels have not been fully completed and clearly understood. To fill this research gap, microstructural evolutions of SLM 17-4 PH steels treated at different aging temperatures and times were carefully characterized to reveal the influence of aging. The corresponding mechanical properties were also studied and analyzed based on microstructure features, especially the precipitation behavior.

## 2. Experimental Details

In this work, spherical 17-4 PH steel powder (Figure 1a) ranging from 30 μm to 40 μm was used as the feedstocks for SLM fabrication. The chemical composition is presented in Table 1. The 17-4 PH steel samples were fabricated using an SLM apparatus of three-dimensional System ProX-200, which was equipped with a fiber laser with a spot diameter of 75 μm, a maximum power of 300 W, and a wavelength of 1070 nm. The SLM fabrication process was conducted in a sealed chamber filled with a protective argon atmosphere (99.99 vol.%). During the SLM, the oxygen content of the atmosphere was controlled to below 200 ppm to avoid oxidation. Prior to the SLM fabrication, the feedstock powders were dried in a vacuum oven at 180 °C for 5 h. The SLM samples were fabricated using the optimal processing parameters to obtain the best relative density (i.e., 99.8 ± 0.12%), as determined by previous researches: a laser power of 200 W, a laser scanning speed of 300 mm/s, a hatch distance of 70 μm, and a layer thickness of 30 μm. Cubic samples with dimensions of 10 mm × 10 mm × 10 mm were fabricated for microhardness measurements and microstructure characterizations. Tensile samples were produced to investigate of the mechanical properties. A scheme of the tensile test specimens is presented in Figure 1b.

Many researchers simply recommend an aging temperature and time of 482 °C and 1 h without providing any reason [28,29,30]. However, it is important to explore the effects of aging parameters on SLM 17-4PH steels. Therefore, different aging temperatures and times ranging from low to high levels were selected for this study. Following SLM manufacturing of the 17-4PH steel samples, direct aging treatments were carried out under various temperatures (from 382 °C to 682 °C) and duration times (from 0.5 to 5 h). These samples are denoted as X °C-Y h, indicating that the samples were aged at X °C for Y h (i.e., 382 °C-2 h representing 382 °C for 2 h).

To determine the phase composition, X-ray diffraction (XRD, Bruker D8 Advance) characterization was performed on the SLM-manufactured 17-4PH steel samples, targeting Cu Kα at a voltage of 40 kV and a current of 35 mA. The XRD spectra were collected by scanning 2θ angles ranging from 30° to 60° with a scanning step size of 0.02°. For microstructure observation, the SLM-manufactured 17-4PH steel samples were polished following the standard metallographic process and then etched with a solution of 5 g FeCl_3_, 20 mL HCl, and 100 mL alcohol. The microstructure evolution of the SLM 17-4PH steel samples was investigated along the cross-sections of the building direction using optical microscopy (OM, OLYMPUS-DP71, Tokyo, Japan) and scanning electron microscopy (SEM, FEI Quenta 450, Portland, OR, USA). The electron-backscattered diffraction (EBSD, TSL-OIM) characterization was conducted on the SLM 17-4PH samples to determine the grain size and crystal orientation. The EBSD scanning was performed at a step size of 70 nm to ensure the adequate resolution of the grains.

Tensile tests of the SLM samples were conducted at a strain rate of 1 mm/min via a universal testing system (INSTRON 5982, Boston, MA, USA). A total of 4 tensile samples were tested in each state of the 17-4 PH steel. The maximum force applied by the universal testing machine Instron 5982 was 100 kN. All sample surfaces before the tests were ground (Ra ≤ 3.2 μm) to avoid measurement errors caused by poor or uneven surfaces. The Vickers microhardness of the SLM-fabricated 17-4 PH steel samples was tested at a load of 500 gf and a dwelling time of 15 s on a microhardness tester (Struers DuranScan 70 G5, Kuchl, Austria). The microhardness values of each sample were averaged over 10 tests. To ensure repitablity and accurancy of the test results, the SLM 17-4PH steel samples before and after different heat treatments were divided into nine groups with five cubic samples for phase detection, microstructural observations, and microhardness measurements, and four tensile samples for mechanical property tests. Please note that only typical experimental results were reported within this study.

## 3. Results and Discussion

### 3.1. Microstructural Evolution in As-Built Condition

Figure 2 presents the microstructure features and phase distribution of the as-built 17-4 PH steel along the building direction. The as-built samples were crack-free, and only a few tiny pores (white arrow) were observed in Figure 2a. Block structures (red arrow) consisting of numerous equiaxed grains (blue arrow) were the main features of the microstructure of as-built samples. Combined with the orientation map and phase distribution map (Figure 2b,c), each packet (purple arrows in Figure 2b) included several blocks that were composed of martensite laths with different misorientations. Moreover, a typical epitaxial growth of the coarse columnar structures (black arrows in Figure 2b) dominated the microstructure of the melt pool boundary of the as-built samples. This solidification mode was driven by the ratio of the thermal gradient (*G*) and cooling rate (*R*). Specifically, a large Δ*G* and a low *R* existed at the molten pool boundaries near the solidified components, leading to the growth of columnar dendrites perpendicular to the boundaries of the molten pool.

Interestingly, a quantity of ultra-fine austenite phase (mainly equiaxed grains in Figure 2c) also formed at the bottom of the molten pools (marked by pink arrows). This phenomenon was most likely due to the intrinsic heat treatment (IHT) of the SLM process. The repetitive re-heating and re-cooling induced by the laser scanning resulted in an IHT effect at temperatures between 300–500 °C, also known as a self-tempering effect [31]. This effect not only promoted the precipitation process of the nano Cu-rich particles [32], but also triggered the generation of the reverted γ-Fe phase (i.e., 3.7 vol.% within the as-built samples). Furthermore, a large fraction of high-angle grain boundaries (HAGBs) (i.e., 66%) was detected in the as-built samples, as presented in Figure 2d, resulting in a large average grain size of the as-built samples (around 10.19 μm). The low-angle grain boundaries (LAGBs) of 34% also indicated a large quantity of dislocations in the as-built 17-4PH steel samples.

### 3.2. Microstructure and Phase Composition in Aging Conditions

Direct aging treatment is an effective method to improve the mechanical properties of martensitic precipitated hardening stainless steel, particularly with regard to temperature and time. Figure 3 presents the effects of aging temperature on the microstructure of the SLM 17-4 PH steel samples. Overall, columnar dendrites dominated the microstructure of the lath-like martensite matrix after aging at different temperatures. Comparatively, the dendrites in the samples aged at low temperatures (Figure 3a,b) were finer than those aged at high temperatures (i.e., Figure 3c,d). The primary dendrites (marked by red arrows in Figure 3c) and secondary dendrites (marked by blue arrows in Figure 3d) were formed with the increasing aging temperature. The essence of aging treatment is a type of tempering process. Therefore, when the 17-4 PH steel was aged at a high temperature (e.g., >500 °C), the lath-like martensite rapidly coarsened, further negatively impacting its mechanical properties.

Moreover, the critical effect of aging treatment is the generation of dispersive nano-Cu particles within the martensite matrix to enhance its material properties [33]. Nanoparticles were scarcely observed within the aged samples at 382 °C for 1 h, while the quantity of precipitated nano-Cu particles (marked by white arrows) swiftly increased with the rising temperature (from 482 °C to 682 °C). Particularly, numerous nano-Cu particles were detected in the aged samples treated at 682 °C for 1 h (Figure 3d). Overall, although the lath width treated at 382 °C was not obviously coarsened, there was no effective precipitation hardening effect (i.e., very few nano precipitates), indicating an under-aging state. After aging above 582 °C, large quantities of nanoparticles were evenly distributed in the matrix, but the lath width was clearly coarsened, biasing towards an over-aging state. Therefore, the 17-4 PH steel, after holding at 482 °C for different times, will be explored in the following section, taking into account the microstructure and the number of precipitated nanoparticles.

To discuss the microstructural evolution during the aging treatment, high-magnification SEM images of the aged samples under different holding times are displayed in Figure 4. Typical lath-like martensite with coarsened secondary dendrites was clearly detected (marked with blue arrows), particularly for long holding times (Figure 4c,d). The strength and hardness of under-aged 17-4 PH steel were restricted despite the fact that the microstructure coarsening caused by short-term aging treatment was not obvious. Long-term aging treatment was beneficial to the precipitation and growth of the nanoparticles, but the coarsened lath martensite greatly reduced its mechanical properties. Therefore, it is extremely significant to seek a desirable combination of the aging temperature and time.

To analyze in detail the reason for under-aging and over-aging phenomena due to the synergic effects of temperature and time, EBSD characterizations were conducted on SLM 17-4 PH steel after different aging treatments. Unlike the coarse packets observed in the as-built samples (Figure 2b), columnar grains with a specific growth direction were replaced by the fine and homogenous martensite laths (Figure 5a). According to the statistics, the average grain size of the SLM 17-4 PH samples was reduced from 10.186 μm (as-builts state) to 3.479 μm (482 °C-1 h aged state), 4.599 μm (482 °C-3 h aged state), and 3.399 μm (582 °C-1 h aged state). Only some molten pools (marked by blue arrows) could be identified within the aged samples treated for 1 h, as displayed in Figure 5a,g, whereas they were difficult to detect in the aged sample designated 482 °C-3 h (Figure 5d). As depicted in Figure 5b,e,h, a similar phenomenon could also be found in the grain boundary distribution statistics. The fraction of the HAGBs within the 482 °C-3 h aged samples (i.e., 76.9%) was higher than that within the short-term aged samples (i.e., 65.2% in the 482 °C-1 h aged samples and 73.7% in the 582 °C-1 h aged samples). LAGBs hinder dislocation movement more significantly than HAGBs [34]. Thus, a higher fraction of LAGBs in the as-built samples (Figure 2d) contributed more strongly to the strengthening mechanism in terms of dislocation movement. However, long maintaining times and high treating temperatures were powerful factors to facilitate dynamic recovery (DRC) and recrystallization (DRX) via the aging processes, which further led to the annihilation of a quantity of subgrain boundaries and dislocations. As such, LAGBs greatly decreased from 34% in the as-built state (Figure 2d) to 23.1% in the 482 °C-3 h aged state (Figure 5e). Generally, as for the 17-4 PH steel, precipitation hardening could become a major strengthening mechanism instead of the working hardening triggered by high-density dislocation walls.

Furthermore, phase mapping was also conducted on the aged samples to reveal the distribution and evolution of the reverted austenite (γ-Fe phase) within the 17-4 PH steel. After aging treatment at 482 °C for 1 h, the volume fraction of the austenite phase increased to 4.5% (Figure 5c) from 3.7% in the as-built sample (Figure 2c). With the prolongation of aging time to 3 h, the austenite phase further increased to 12.0% (Figure 5f). A high aging temperature of 582 °C for 1 h also resulted in an increase of the austenite phase to 8.2% (Figure 5i), compared with the aged sample at 482 °C for 1 h. Interestingly, the reverted austenite (i.e., γ-Fe phase) was concentratedly formed at the interface of the molten pools (marked by red arrows) after the aging treatments, as presented in Figure 5c,i. This was most likely due to the aging treatments contributing to the promotion of growth of the in situ generated γ-Fe phase around the molten pools. Notably, compared to other aged samples (Figure 5c,i), aging time was a critical factor to boost the formation of the γ-Fe phase (i.e., 12.0% in Figure 5f), leading to good ductility. Therefore, balancing the aging temperature and holding time was important to obtain excellent mechanical properties in the 17-4 PH steel.

To further identify the phase distribution, XRD analysis was conducted on the SLM 17-4 PH steel under different aging states. As shown in Figure 6, the dominant phase compositions of the aged samples were α’-Fe phase (JCPDS # 06-0696) and γ-Fe phase (JCPDS # 23-0298). Figure 2c shows that the SLM 17-4 PH samples in the as-built state were mainly composed of the α’-Fe phase with a BCC structure and a small amount of γ-Fe phase with a FCC structure. Notably, the γ-Fe phase within the as-built samples did not exclusively include the retained austenite (i.e., γ-Fe phase) due to an incomplete transformation of austenite to martensite at a high temperature. The samples also included reverted austenite (named γ’-Fe phase to show this distinction) caused by the IHT effect during the SLM. After being processed by the aging treatment at various temperatures (Figure 6a) and times (Figure 6b), the reverted austenite with an FCC structure was rapidly formed alongside numerous nano-Cu particles. In short, phase transition within the aged samples can be described as follows:$$\alpha^{\prime }+\gamma +\gamma^{\prime }\overset{\mathrm{aging}\;\mathrm{treatment}}{\to }\alpha^{\prime }+\gamma^{\prime }+\mathrm{nano}-\mathrm{Cu}\;\mathrm{particles}$$

The types of austenite (i.e., retained austenite and reverted austenite) could not be precisely distinguished. Thus, only ‘γ-Fe phase’ will be used throughout the remainder of this manuscript to improve readability. To quantitatively determine the amount of γ-Fe phase in the aged samples, a Reference Intensity Ratio (RIR) and Rietveld Refinement analysis were conducted on the XRD spectrum of the SLM 17-4 PH steel samples under different aging conditions. The ‘error’ in Table 2 represents the difference between the fitted curves and the XRD spectra; i.e., fitting curve error. As displayed in Table 2, the volume fraction of the γ-Fe phase increased with further aging treatment, which agreed with the EBSD phase mapping results (i.e., Figure 5c,f,i). This increase in austenite content at higher aging temperatures and longer duration complied with the phenomena reported in the previous section. Notably, the aging time was more vital than the holding temperature in determining the outcome of long-term aging (i.e., 19.6%). The formation of austenite grains through reversion from the martensite phase was most likely to occur at the boundaries of martensite laths during elongated aging times and elevated temperatures. Reverted austenite is beneficial to the chemical stabilization process by diffusing alloying element constituents and promoting ductility [35]. Nonetheless, the high-content γ-Fe phase was accompanied by Cu-rich dispersoids (above 550 °C [17]), reducing its tensile strength.

In summary, the microstructure of the aged 17-4 PH steel frequently exhibited three states, owing to the different treatment temperatures and holding times, as shown in Figure 7. As depicted in the as-built samples (Figure 7a), only a small amount of γ-Fe phase (i.e., primarily in the form of retained austenite) was generated within the interface of the molten pools. A quantity of ultra-fine columnar grains (blue wireframes) and equiaxed grains (yellow hexagons) constituted the martensite packets (i.e., HAGBs) with numerous dislocations (i.e., LAGBs). Only a small amount of γ/γ’-Fe phase (pink ellipses) concentratedly grew at the interface of the molten pools due to the IHT effect. After an aging treatment with a low temperature or a short holding time, the first type of aging state, an under-aged state, is presented in Figure 7b. A few nano-Cu particles (green spots) were in situ precipitated within the martensite matrix, while coarse microstructures (i.e., coarsened columnar grains and elongated equiaxed grains) were also inevitably formed owing to the DRC and DRX process. Although in situ-formed nano-Cu particles boosted the hardness and tensile strength of the SLM 17-4 PH steel, the mediocre mechanical properties were not suitable for some load-bearing applications. Thus, to produce more nano-Cu particles that further improved the mechanical properties of the SLM 17-4 PH steel, a high aging temperature and long holding time were conducted on the as-built samples. As presented in Figure 7c, numerous dislocations were eliminated due to the DRC/X effects, thus leading to the coarse laths-like martensite and elongated equiaxed grains rather than packets. Not only quantities of nano-Cu particles, but also γ-Fe phase (mainly reverted austenite) were formed within the matrix. Thus, the mechanical properties were greatly improved due to the precipitation hardening aroused by the Cu-rich nanoparticles within the adequate-aged state instead of the dislocation strengthening within the as-builts state. However, with the further increase in the aging temperature or holding time, as displayed in Figure 7d, nano-Cu particles (marked by green spots) and microstructures were extremely coarsened, resulting in a great reduction of the mechanical performance; i.e., an over-aged state. The strength and hardness of the over-aged samples was significantly weakened due to the unfavorable microstructural features. In short, different aging conditions caused distinctive mechanical properties. Understanding the microstructural evolution mechanisms and seeking a series of optimal aging parameters are essential in this work.

### 3.3. Mechanical Properties

Figure 8 shows the microhardness variation of the SLM-manufactured 17-4 PH steel samples after different aging temperatures and holding times. As displayed in Figure 8a, with the prolongation of aging temperatures, the microhardness first increased rapidly and then slowly decreased from an under-aged state to an over-aged state. The microhardness peak (430 ± 8.5 HV_0.5_) appeared in the 482 °C-1 h aged samples. The relationship between the microhardness (HV, HV_0.5_) and the aging temperature (T, °C) can be depicted as follows:$$\mathrm{HV}=337.88+0.33T-3.09\times 10^{-4}T^{2}$$

As for the effect of the aging time on the microhardness of the aged samples (Figure 8b), the microhardness also first increased, then remained roughly unchanged. The maximum value (455 ± 8.9 HV_0_._5_) occured within the 482 °C-3 h aged samples. The function of microhardness (HV, HV_0_._5_) vs. aging time (t, h) can be concluded as follows:$$\mathrm{HV}=-113.07\times e^{-\frac{t}{0.59}}+451.26$$

Thus, the optimal aging parameters suitable for SLM 17-4 PH steel can be selected as 482 °C for 3 h. Please note that the aforementioed equations only display the propbable relationship between the microhardness and aging temperature alongside the aging time. These functions cannot show the strict quantitative relationship between these factors due to limited testing results.

Figure 9a,b illustrates the strain-stress curves of the SLM 17-4 PH steel samples under different aging conditions, and the variations of tensile performances are summarized in Figure 9c,d, including ultimate tensile strength (UTS), yield strength (YS), and elongation to break (EL). As can be seen in Figure 9a,c, both the UTS and YS gradually increased withthe prolonged aging temperature at 1 h aging time. The UTS reached its peak value at the aging temperature of 482 °C, which was similar to the trend in microhardness. The ductility decreased rapidly (i.e., EL = 8.8%) when the aging treatment was conducted on the SLM 17-4 PH steel at 482 °C, despite having the highest strength (i.e., YS = 939.7 MPa and UTS = 1312.8 MPa) among the tested steel samples. Thus, the aging temperature should preferably be 482 °C if the strength is the optimization goal.

The results of further optimization for the aging temperature of 482 °C are shown in Figure 9b,d. Strength values, including the YS and UTS, increased with the prolongation of the aging time from 0.5 h to 5 h (Figure 9b). The highest strength I when the aging time was 3 h, indicating an adequate aging state that offered an important positive effect. Although EL was not the best among these aged samples, ultra-high UTS (i.e., 1353.4 MPa) with a desirable ductility (i.e., EL = 10.8%) displayed an excellent combination of strength and toughness under this aged state, as the statistics show in Figure 9d. Interestingly, YS and EL exhibited a slight elevation compared to the other aged states, while an extreme reduction of the UTS was shown. These effects could be attributed to the over-aging treatment, resulting in an obvious coarsening phenomenon of the nanoparticles and the microstructure. Thus, the fracture would occur immediately once the yield point was exceeded. In summary, the comprehensive properties of the 482 °C-3 h aged samples can be considered the best among these aged samples.

Moreover, the elongation value exhibited an inverse tendency compared with the tensile strength. Based on the above mentioned results, increases in aging time and temperature both initially increased the tensile strength of SLM 17-4 PH steel samples, followed by a subsequent decrease. The microstructure observation in Figure 5 shows that shorter aging times and lower aging temperatures both resulted in fewer nanoprecipitates. Thus, it is reasonable to conclude that higher aging temperatures and longer aging times resulted in a more significant strengthening effect, leading to higher strength values. On the contrary, further increasing the aging temperature and time can result in an increased fraction of the austenite phase and a coarsened microstructure, reducing the strengthening effect and lowering the strength values.

Figure 10 presents the fracture surfaces of the SLM 17-4 PH steel samples under different conditions, revealing the typical effect of aging. Compared with other aged states, only the as-built samples had a clear necking phenomenon (red arrows), as displayed in Figure 10a. Partially unmelted powder particles (blue arrows) and cracks (green arrows) could be also detected. The uniform distribution of the numerous micro-sized dimples (pink arrows) found using a high-magnification SEM shows that a ductile fracture behavior dominated the as-built samples. After an under-aging treatment (e.g., 482 °C for 1 h), Figure 10b shows the presence of not only small dimples but also many river patterns (purple arrows), indicating a mixed fracture mode of brittleness/ductility after the aging treatment. With an increase in the holding time, small dimples and obvious cleavage facets were observed on the fracture surface of the 482 °C-3 h aged samples, as displayed in Figure 10c. This phenomenon indicates that a large quantity of precipitated nano-Cu particles resulted in a brittle fracture, which is consistent with the trend in strength/ductility of tensile performance. When the aging time was further increased, an over-aged state was characterized, as shown in Figure 10d. Notably, tear ridges were formed along the interface of the martensite laths, and many dimples suggested poor ductility. As for the precipitation hardening alloys, the existence of secondary nanoparticles and precipitates could initiate the nucleation of micro-sized voids during the tensile fracture. Thus, the growth and coalescence of voids would result in the observed dimple features on the fracture surface in the tensile loading process. With a prolonged aging time, more prominent precipitation of secondary nanoparticles could promote the generation of more voids, leading to an increased number of dimples with small diameters.

## 4. Conclusions

Despite an ultra-high strength in the as-built condition, the post-aging treatment is still indispensable for SLM 17-4 PH steel to obtain balanced strength and ductility. This work investigated the effects of aging temperature and time on the microstructure and mechanical properties of SLM 17-4 PH steel. The main conclusions of this work are summarized as follows:In the as-built conditions, the SLM 17-4 PH steel exhibited a dense microstructure with no significant pores and cracks. The microstructure of the as-built samples exhibited columnar grains growing along the building direction as a result of the largest temperature gradient in the SLM process. Ultra-fine equiaxed grains were decorated at the molten pool boundary, which was confirmed as γ-Fe phase.Based on SEM and EBSD observations, coarse martensite laths were observed in the aged samples compared with the as-built ones regardless of the aging time and temperature. Increasing the aging temperature led to more significant grain growth of the martensite laths.In the as-built condition, the 17-4 PH steel sampless were mainly composed of the martensite phase with a BCC structure and a small amount of austenite phase with a FCC structure. The aging treatment at various temperatures and times led to the formation of the reverted austenite phase with a FCC structure. As the aging time increased at 482 °C, the SLM 17-4 PH steel exhibited an increasing intensity of the austenite peak in comparison with the as-built samples. With further aging treatment, the increasing volume fraction of the austenite phase was found, which was consistent with the EBSD phase mapping results.The mechanical properties showed that both the UTS and yield strength gradually increased as the aging time at 482 °C was prolonged. The UTS reached its peak value at an aging time of 3 h and a temperature of 482 °C, which was similar to the trend in microhardness. However, the ductility decreased rapidly with the aging treatment of SLM 17-4 PH steel.

## Figures and Tables

**Figure 1 micromachines-14-00871-f001:**
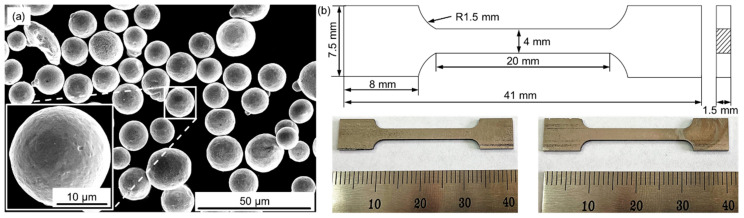
(**a**) Surface morphology of the 17-4PH steel powder; (**b**) scheme of the tensile test specimens used for the SLM fabrication.

**Figure 2 micromachines-14-00871-f002:**
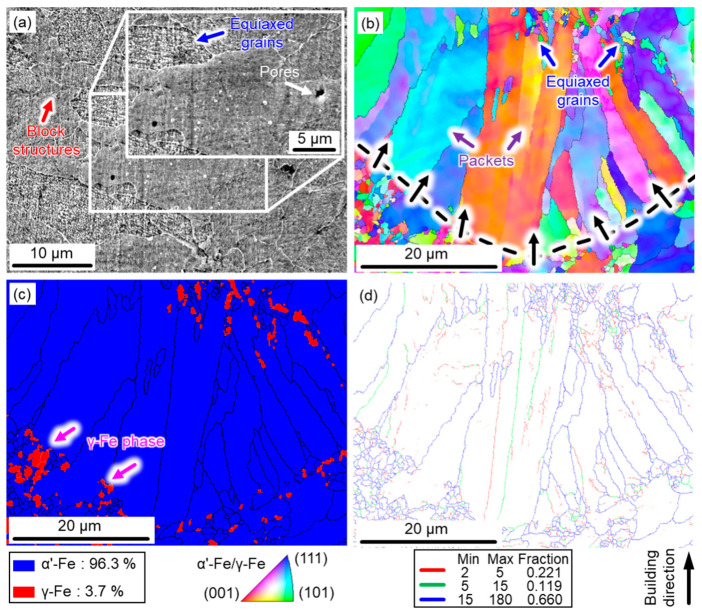
Microstructures of the SLM 17-4 PH steel under as-built conditions: (**a**) SEM; (**b**) IPF map; (**c**) phase distribution; (**d**) grain boundary map.

**Figure 3 micromachines-14-00871-f003:**
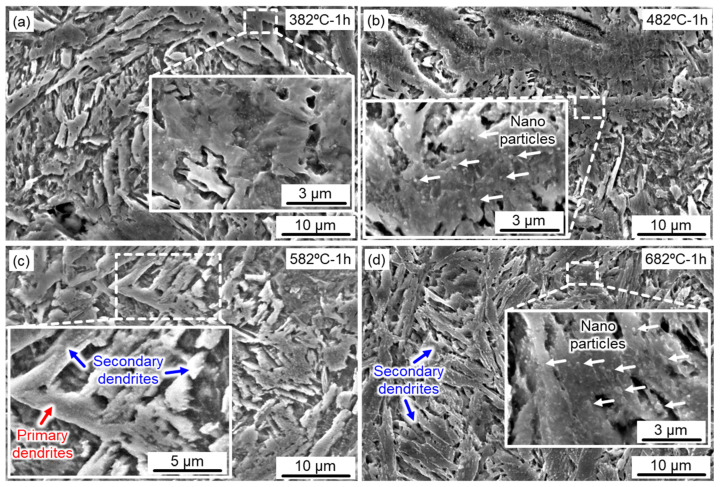
Microstructural evolution of the aged SLM 17-4 PH steel at various aging temperatures for 1 h: (**a**) 382 °C; (**b**) 482 °C; (**c**) 582 °C; (**d**) 682 °C.

**Figure 4 micromachines-14-00871-f004:**
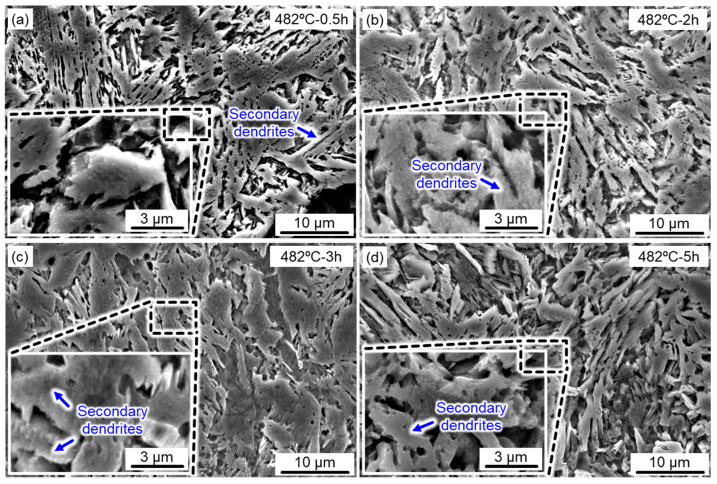
Microstructure features of the aged SLM 17-4 steel at 482 °C for different holding times: (**a**) 0.5 h; (**b**) 2 h; (**c**) 3 h; (**d**) 5 h.

**Figure 5 micromachines-14-00871-f005:**
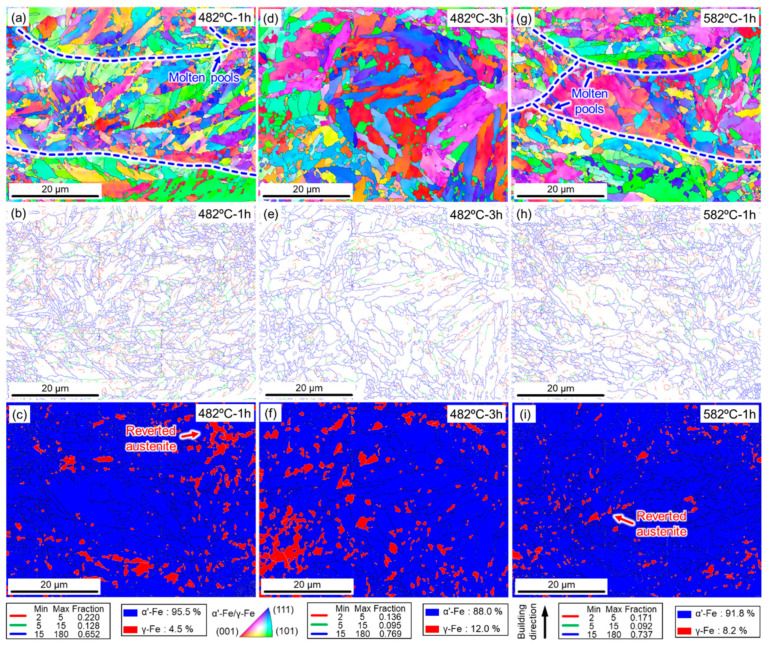
EBSD characterizations of the SLM 17-4 PH steel samples under different conditions: (**a**–**c**) aged samples at 482 °C for 1 h; (**d**–**f**) aged at 482 °C for 3 h; (**g**–**i**) aged at 582 °C for 1 h. IPF images are shown in the first row, grain boundary distribution in the second row, and phase mapping in the third row.

**Figure 6 micromachines-14-00871-f006:**
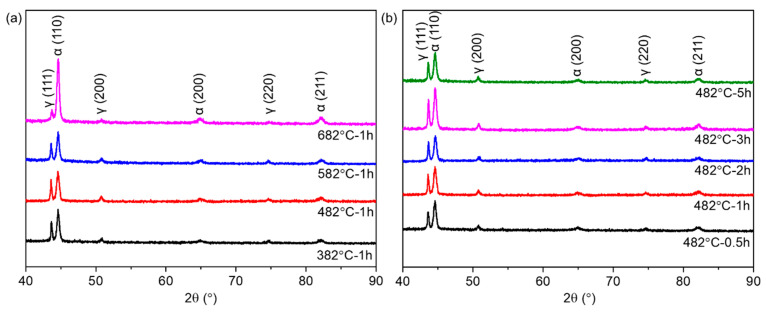
XRD spectra of the SLM-manufactured 17-4 PH steel samples under different (**a**) aging temperatures and (**b**) aging times.

**Figure 7 micromachines-14-00871-f007:**
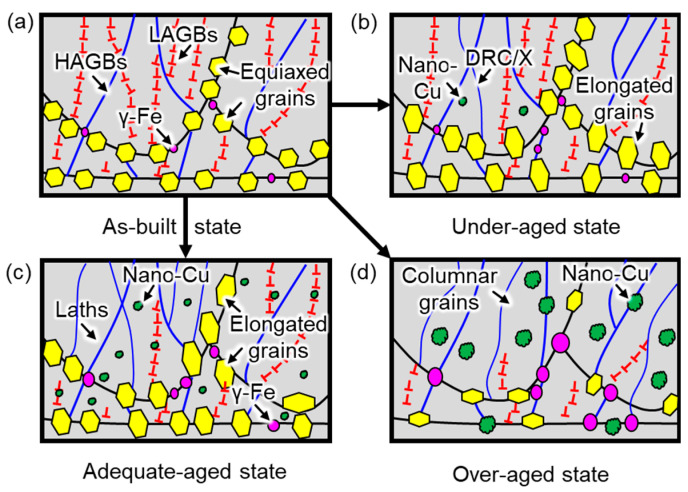
Typical microstructural evolution of the different aged states: (**a**) as-built; (**b**) under-aged; (**c**) adequate-aged; (**d**) over-aged.

**Figure 8 micromachines-14-00871-f008:**
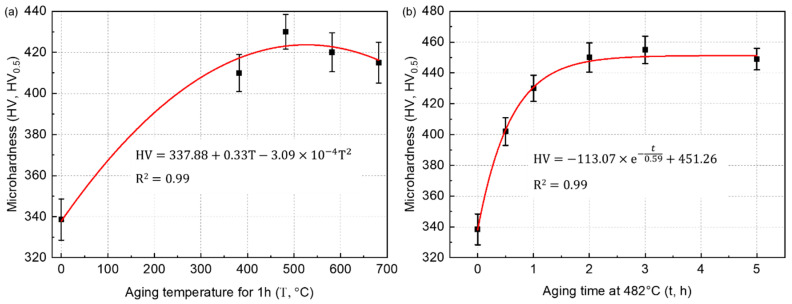
Microhardness variation of the SLM-manufactured 17-4 PH steel samples after aging treatments under different (**a**) temperatures and (**b**) holding times.

**Figure 9 micromachines-14-00871-f009:**
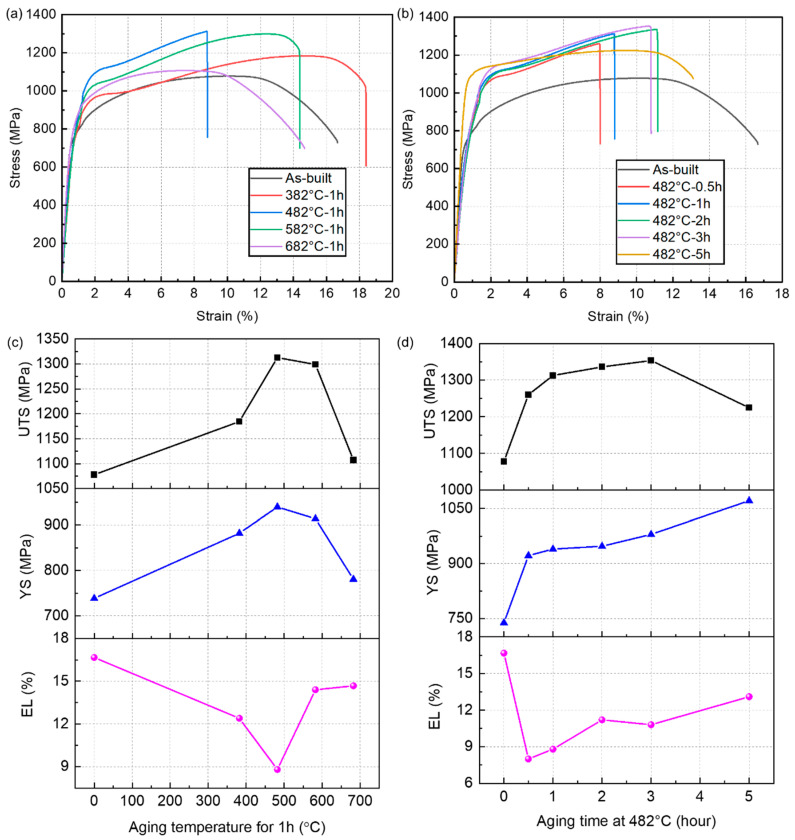
Mechanical properties of the SLM 17-4 PH steel samples under different post-aging treatment conditions: (**a**,**b**) strain-stress curves and (**c**,**d**) key indicator statistics. First row: mechanical properties at different aging temperatures for 1 h. Second row: mechanical properties at different aging times at 482 °C.

**Figure 10 micromachines-14-00871-f010:**
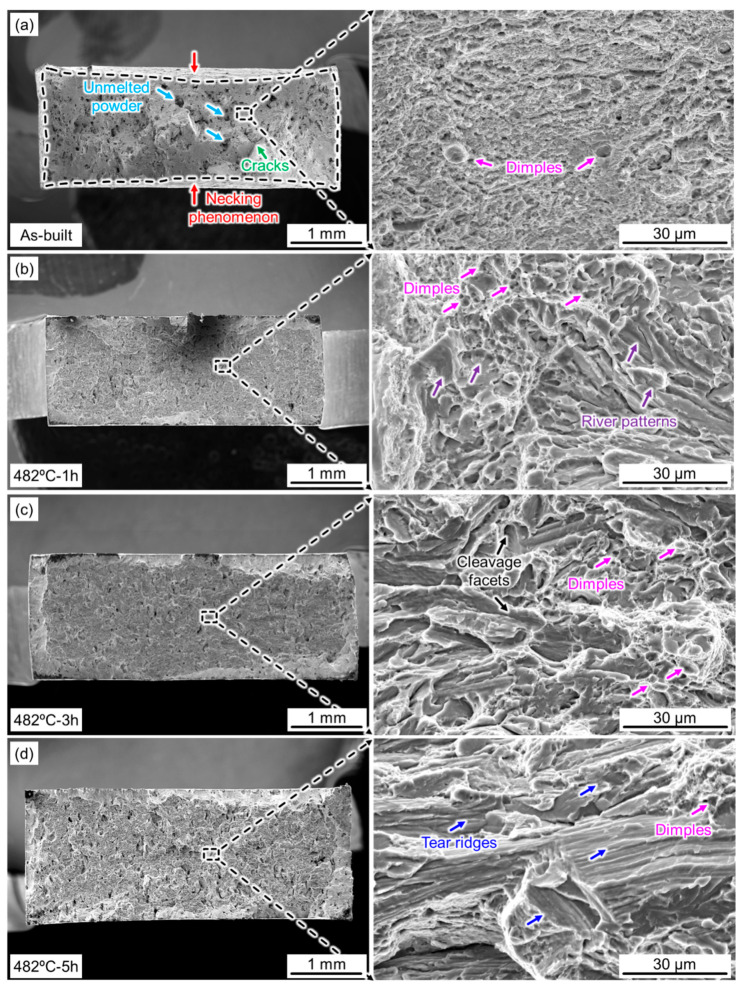
Fractographic features of the SLM 17-4 PH steel samples under different conditions: (**a**) as-built; (**b**) aged at 482 °C for 1 h; (**c**) aged at 482 °C for 3 h; (**d**) aged at 482 °C for 3 h.

**Table 1 micromachines-14-00871-t001:** Chemical composition of the 17-4 PH steel powder (provided by EOS GmbH).

Elements	C	Cr	Ni	Cu	Mo	Nb	Mn	Si	P	S	Fe
Weight percentage (wt.%)	≤0.07	15.0–17.0	3.0–5.0	3.0–5.0	≤0.6	≤0.45	≤1.5	≤0.7	≤0.04	≤0.03	Bal.

**Table 2 micromachines-14-00871-t002:** Volume fractions of the α’-Fe and γ-Fe phases in the SLM 17-4 PH steel samples under different states.

Samples	α’-Fe Phase (%)	γ-Fe Phase (%)	Fitting Curve Error
As-built	96.3	3.7	/
Aged at 382 °C for 1 h	95.1	4.9	3.5
Aged at 482 °C for 1 h	93.8	6.2	4.2
Aged at 582 °C for 1 h	89.7	10.3	3.3
Aged at 682 °C for 1 h	80.4	19.6	2.1
Aged at 482 °C for 0.5 h	94.6	5.4	2.2
Aged at 482 °C for 2 h	88.9	11.1	1.8
Aged at 482 °C for 3 h	86.4	13.6	3.5
Aged at 482 °C for 5 h	83.7	16.3	2.6

## Data Availability

Not applicable.

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
