# Peer review of "Influence of Aging Treatment Regimes on Microstructure and Mechanical Properties of Selective Laser Melted 17-4 PH Steel"

_micromachines, 2023, doi:10.3390/mi14040871_

Round 1
Reviewer 1 Report
The manuscript entitled “micromachines-2297148” dealing with laser processing (SLM) has been reviewed. The paper has been nicely written but needs significant improvement. Please follow my comments.
1. What is the main issue that will be solved by this investigation? Please clarify it in the text.
2. Please add a brief statement on your methodology in the abstract.
3. More detail about “Figure 2. Microstructures of the SLM 17-4 PH steel under as-built condition: (a) SEM; (b) IPF map; (c) phase distribution; (d) grain boundary map.” Is needed.
4. What is the future direction of this work?
5. Laser absorptivity is important which shows the quality of the parts and transition from keyhole to conduction mode. Please read and add the following ref in this area. “The effect of absorption ratio on meltpool features in laser-based powder bed fusion of IN718”.
6. Please proofread the paper.
7. Laser has many advantages over the conventional manufacturing method which can be highlighted in your paper. Please read the following manuscript and add it to the literature to show how the laser is comparable with conventional manufacturing.
· Laser subtractive and laser powder bed fusion of metals: review of process and production features
Author Response
Thank you for your comments, please see the attachment.

Reviewer 2 Report
· The level of technical English is very low. A reader who is not very familiar with the topic of the article may misinterpret a large part of the technical data and explanations due to poor translation. The reviewer had difficulties understanding some parts of the text because some descriptions and especially some parts of the results analysis is not clearly presented.
· The material and methods part is superficial and does not present clear information of the conducted research.
· The authors state that the specimens were built in SLM using the optimal processing parameters. Optimal processing parameters for what? To obtain full density? To obtain specified mechanical properties or specified hardness and microstructure? Who states that those are optimal parameters? Manufacturer of the SLM equipment? Previous research?
· Aging treatment was conducted where, which furnace, with or without protective atmosphere, which were the correct parameters (time, temperature, heating/cooling speed). Why were specified various temperatures (form 382 °C to 682 °C) chosen, and not 350 – 700 e.g.? Why 382 and not 380?
· The clear presentation of heat treatments and specimen markings/groups is missing. How many specimen groups were prepared and why?
· Scheme of tensile test specimens with correct measurements is missing. Also, the tensile test specimen’s measures are chosen form where, are they standard?
· Why is the indentation time 25 s in Vickers microhardness test, usual times for ferrous metals is 10 – 15 s?
· The microstructural analysis is hectic because the clear connection between aging treatment and specimen group/mark is missing.
· In microhardness measurements, how were the equations displaying the relationship between the microhardness and the aging temperature and the time obtained? Statistical analysis, how many data were used to generate those equations? Was some factorial design or analysis of variance used?
Round 2
Reviewer 1 Report
The paper is ready to publish.
Author Response
The reviewer did not have further comments.
Reviewer 2 Report
Authors addressed some reviewers' comments in good manner except:
· Scheme of tensile test specimens with correct measures is missing. Also the information on maximum force of the universal testing machine Instron 5982. Authors write: "Tensile samples (gauge size of 25 mm × 3 mm × 3 mm) were produced for the investigation of the mechanical properties." and later "Tensile tests of the SLM samples were conducted at a strain rate of 1 mm/min via a universal testing system (INSTRON 5982, USA) based on the ASTM E8/E8M-16a stand-9 ard, and a total of 4 tensile samples were tested in each 17-4 PH steel state. Derailed measuring descriptions can be found in our previous studies [16]." The cited work does not provide sufficient information and does not answer the question of missing scheme of tensile test specimens and why this shape was chosen. What is the length of the reduced section? What is the width of the grip? According to the measurements, are the specimens square-shaped 3 x 3 mm? Were the specimens treated for surface finish somehow because 3x3 mm square shape produced in SLM could have surface roughness that could affect the results of the tensile test. Also, figure 10 presents fractographic surfaces of the tensile samples. From the figures it can be seen that the shape of the specimens is not square 3x3 mm but rectangular approx. 4 x 2 mm or similar??
· Why is the indentation time 25 s in Vickers microhardness test, usual times for ferrous metals is 10 – 15 s?
· In microhardness measurements, how were the equations displaying the relationship between the microhardness and the aging temperature and the time obtained? Statistical analysis, was some factorial design or analysis of variance used?
Author Response
The part of tensile specimen has been revised
Round 3
Reviewer 2 Report
Authors have improved the paper but still some information on testing methods used for mechanical properties are inconclusive.
The information on hardness testing are missing. Which microhardness testing machine was used and why is the 25 s indentation time chosen. ASTM 384 Standard Test Method for Microindentation Hardness of Materials specifies: 6.1.1.3 The full test force shall be applied for 10 to 15 s unless otherwise specified. 6.1.1.4 For some applications it may be necessary to apply the test force for longer times. In these instances the tolerance for the time of the applied force is +/- 2 s.
Please elaborate.
Again, in microhardness measurements, how were the equations displaying the relationship between the microhardness and the aging temperature and the time obtained? Statistical analysis, was some factorial design or analysis of variance used? Please specify .
In tensile test, why is the reference 16 chosen for detailed information when in this paper is less information on tensile test than in this.
The authors added force on tensile machine, maximum force 100 kN. For specimens 4 x 1,5 mm with the reduced section length 20 mm and grip dimensions 8 x 7,5 x 1,5 mm how was the stress-strain data obtained and secured? By extensometer or some other device because on 100 kN tensile machine the measurement uncertainty can have significance influence on the results for specimens with this dimension.
